1.5em

R E S C I E N C E C

Replication Study

# [Re] BiRT: Bio-inspired Replay in Vision Transformers for Continual Learning

***[1], ID

[1] Purdue University, West Lafayette, Indiana

**Edited by**
(Editor)

**Reviewed by**
(Reviewer 1)
(Reviewer 2)

**Received**
08 November 2023

**Published**
–

**DOI**
–

## 1 Replication Summary

The base code for ViT model used in this replication study which was introduced in "An image is worth 16x16 words: Transformers for image recognition at scale" [1] is similar to [2]. The code implementation for this replication study for all the ideas introduced in the paper including memory updation algorithm, memory structure, and training architecture is strictly original and has not been taken from any other source. The experiments were done for 500 epochs. The replication of this study was constrained by computational limitations, specifically in terms of time. Due to the absence of a mechanism for storing memory to be reused once the computation time limit was reached as of yet (time of submission of this paper), the original experiment could not be fully replicated. Instead, a modified version of the study was conducted with a reduced number of epochs, and the summarized results are presented in the paper

## 2 Analysis of the Original Paper

1. The authors failed to provide exact values of several hyper parameters and parameters used in their equations described in the proposed method section of their paper [3]. Thus, there is no way to replicate the exact same experimental set up.

   - They failed to provide information for the hyper-parameters $\alpha_t, \alpha_m, \alpha_a, \alpha_s$, and the ones in equation 3, 4, 6, 8 used in their experiments specific to their architecture.
   - They have also not provided any information about the pre-processing steps taken for their dataset.

2. The proposed method focuses on continual learning, where the model learns sequentially from disjoint datasets representing different tasks. The paper describes a fine-tuning strategy on a balanced dataset after each task for 20 epochs. However, it is noted that the majority of learning occurs in the initial epochs. For instance, training the Vision Transformer (ViT) model on datasets like CiFAR10 and CiFAR100 for just 5 epochs already yields a notable accuracy of 45% and 18% respectively. This observation questions the necessity of 20 epochs of fine-tuning after each task. Fine-tuning on a balanced dataset containing samples from all tasks after each task is undermines the essence of continual learning.

# 3  Dataset and Data processing

The dataset used for training and testing purposes was CiFAR10 and CiFAR100.

- **CiFAR10**: It consists of 60,000 32x32 color images in 10 different classes, with 6,000 images per class. The dataset is split into 50,000 training images and 10,000 testing images.

- **CiFAR100**: It consists of 60,000 32x32 color images in 100 different classes, with 600 images per class. The dataset is split into 50,000 training images and 10,000 testing images.

The following transformations were applied to the dataset in the given sequence to augment the data and make it more robust to variations in the input.

1. **transforms.ToTensor():**

    - Converts the input image to a PyTorch tensor. Changes the image data type from a PIL Image or numpy array to a PyTorch tensor.

2. **transforms.Resize((32, 32)):**

    - Resizes the input image to a fixed size of 32x32 pixels.

3. **transforms.RandomHorizontalFlip(p=0.5):**

    - Randomly flips the input image horizontally with a probability of 0.5. Introduces a form of data augmentation by providing different views of the same image.

4. **transforms.RandomResizedCrop((32, 32),scale=(0.8, 1.0), ratio=(0.75, 1.33), interpolation=2)**

    - Randomly crops and resizes the input image. The scale parameter controls the range of the cropped area as a ratio of the original image size

    - The ratio parameter controls the aspect ratio of the cropped area. Interpolation=2 specifies bilinear interpolation for resizing.

5. **transforms.Normalize((0.5, 0.5, 0.5), (0.5, 0.5, 0.5)):**

    - Normalizes the pixel values of the input image. Subtracts the mean and divides by the standard deviation. Assumes the image has three color channels (e.g., RGB) and standardizes the pixel values to be in the range [-1, 1]. Mean and standard deviation values used are both set to 0.5.

Since the paper primarily focuses on Continual Learning, the dataset needed to be prepared for continual learning. Continual learning refers to the ability of a machine learning model to incrementally acquire and adapt to new information over time, without requiring retraining on the entire dataset. It enables the model to learn from a stream of data in a dynamic environment, allowing it to retain knowledge from past experiences while incorporating new knowledge efficiently.
This was done by separating the dataset with respect to each class and then creating subsets. The number of subsets was equal to the number of tasks. The classes were separated into tasks while ensuring that all tasks remained disjoint. The model is then trained on each of these tasks successively and fine-tuned on a balanced dataset at the end of each task.

## 4 Key Ideas Introduced in the Original Paper

The continual learning paradigm normally consists of $T$ sequential tasks, with the data gradually becoming available over time. During each task $t \in \{1, 2, \ldots, T\}$, the samples and the corresponding labels $(x_i, y_i)_{i=1}^N$ are drawn from the task-specific distribution $D_t$.

### 4.1 Knowledge consolidation through complementary learning system

Complementary learning system posits that the hippocampus and neocortex entail complementary properties necessary to capture complex interactions in the brain [4]. Inspired by CLS they propose a dual memory transformer-based learning system in which the working model encounters new tasks and consolidates knowledge over short periods of time which is then gradually aggregated into the weights of the semantic memory during intermittent stages of inactivity.

$$\theta_s = \gamma\theta_s + (1 - \gamma)\theta_w \tag{1}$$

### 4.2 Episodic Memory

They propose a high level representation rehearsal for vision transformers. The working model comprises two nested functions: $g()$ and $f_w()$. The first few layers of the encoder $g()$, processes the raw image input, and the output along with the ground truth label is stored in the episodic memory $D_m$, $f_w()$ and its stable counter part $f_s()$ is updated according to equation 1. They populate the episodic memory at the task boundary using iCarL herding [5] at the end of task boundry. The algorithm used to implement iCarl herding is described in the next section. The learning objective for representation rehearsal is given in equation 2

$$L_{er} = E_{(x_i, y_i) \sim D_t}[L_{ce}(f_\theta(x_i), y_i)] + \alpha E_{(x_j, y_j) \sim D_m}[\mathcal{L}_{ce}(f_\theta(x_j), y_j)] \tag{2}$$

### 4.3 Noise and Trial-to-Trial Variability

Noise is prevalent at every level of the nervous system and has been shown to play constructive role in brain. Furthermore, injecting noise into the neural network learning pipeline has been shown to result in faster convergence to the global optimum [6] , better generalization [7], and effective knowledge distillation.

**Representation Noise $\widetilde{M}$ −**

$$\widetilde{r} = \lambda\mathbf{r}_i + (1 - \lambda)\mathbf{r}_j \tag{3}$$

$$\widetilde{y} = \lambda\mathbf{y}_i + (1 - \lambda)\mathbf{y}_j \tag{4}$$

The authors propose to linearly combine the representations sampled from episodic memory using a manifold mixup as shown in **??** where $r_i$ and $r_j$ are stored representations of two different samples, and $y_i$ and $y_j$ are the corresponding labels.
This concept has not been explored in this implementation.

**Attention Noise $\widetilde{A}$ −** The working model $f_w(.)$ in BiRT consists of several multi-head self-attention layers that map a query and a set of key-value pairs to an output. The authors inject noise into the scaled dot-product attention at each layer of $f_w(.)$ while replaying the representation as shown in equation 5.

$$\text{Attention}(Q, K, V) = \left( \text{softmax}\left( \frac{QK^T}{\sqrt{d_k}} \right) + \epsilon \right) V \tag{5}$$

where $Q$, $K$, and $V$ are query, key, and value matrices, and $\epsilon \sim \mathcal{N}(0, \sigma^2)$ is a white Gaussian noise. By stochastically injecting noise into self-attention, they discourage BiRT from overfitting.

**Label Noise $\widetilde{T}$ –** The author introduce a synthetic label noise $\widetilde{T}$ in which they re-assign a small percentage of the samples a random class, thus taking advantage of the fact that label noise is sparse in the real world [8].

**Supervision Noise $\widetilde{S}$ –** They also regularize the function learned by the working model to enforcing consistency in its predictions with respect to the semantic memory using equations 6 and 7.

$$L_{cr} = \beta_1 E_{x_i \sim D_t} \| f_w(g(x_i)) - f_s(g(x_i)) \|_p + \beta_2 E_{r_j \sim D_m} \| f_w(r_j) - f_s(r_j) \|_p, \tag{6}$$

$$f_s(r_j) \leftarrow f_s(r_j) + \delta, \tag{7}$$

where $\beta_1$ and $\beta_2$ are balancing weights, $\delta \sim \mathcal{N}(0, \sigma^2)$ is a white Gaussian noise, and $L_{cr}$ represents the expected Minkowski distance between the corresponding pairs of predictions, and $p = 2$. Thus the final learning objective becomes equation 8

$$L = L_{\text{repr}} + \rho L_{\text{cr}} \tag{8}$$

## 4.4 Algorithm Used by the Authors

The algorithm used by the authors [3] is presented in algorithm 1.

## 4.5 Models and Algorithms Used in this Implementation

**Vision Transformer and Working Model Architecture $f_w()/f_s()$ and $g()$ –** The Vision Transformer Architecture is largely taken and is similiar to the original paper "An image is worth 16x16 words: Transformers for image recognition at scale" [1] [2]. The paper introduces an input embedding layer, followed by multiple encoder blocks followed by a MLP head. The architecture is summed up in the image 1. The values of the parameters used are summarized in tables 1,2. The $g()$ model contains the embedding layer and 2 encoder blocks from the encoder stack. The input to this model is of the dimension batch * channel * height * width and the output is batch * patches + 1* hidden_size. The $f_s()$ and $f_w()$ consist of num_heads - 2 encoder blocks and the MLP head. The input to this model is of the dimension batch * patches + 1 * latent size and the output is the batch * 1.

---

**Algorithm 1** BiRT Algorithm

---

**input:** Data streams $\mathcal{D}_t$, buffer $\mathcal{D}_m$, working model $f_w$, hyperparameters $\gamma$, $\alpha_t$, $\alpha_m$, $\alpha_a$, $\alpha_s$

**for** tasks $t \in \{1, 2, .., T\}$ **do**
    **for** epochs $e \in \{1, 2, .., E\}$ **do**
        Sample a mini-batch $(x, y) \sim \mathcal{D}_t$
        $x = \text{augment}(x)$
        **if** $\mathcal{D}_m \neq \emptyset$ **then**
            Sample a mini-batch $(r, y) \sim \mathcal{D}_m$
            $a, b, c, d, e \sim \mathcal{U}(0, 1)$
            $\tilde{y} \leftarrow \tilde{\mathcal{T}}(y)$         **if** $a < \alpha_t$
            $(\tilde{r}, \tilde{y}) \leftarrow \tilde{\mathcal{M}}(r, y)$    **if** $b < \alpha_m$                 $\triangleright$ (Eq. 3, 4)
            $\tilde{A} \leftarrow \tilde{\mathcal{A}}(A)$         **if** $c < \alpha_a$                   $\triangleright$ (Eq. 5)
            $f_s(r) \leftarrow \tilde{\mathcal{S}}(f_s(r), \delta)$ **if** $d < \alpha_s$       $\triangleright$ (Eq. 7)
        **end if**
        Compute outputs of $f_w(.)$ and $f_s(.)$
        Compute $\mathcal{L} = \mathcal{L}_{repr} + \rho\mathcal{L}_{cr}$                 $\triangleright$ (Eqs. 2, 6, 8)
        $\theta_w \leftarrow \theta_w + \nabla_{\theta_w}\mathcal{L}$
        $\theta_s \leftarrow \gamma\theta_s + (1 - \gamma)\theta_w$ **if** $e < \alpha_e$ and $t > 1$
    **end for**
    **if** task-end = True **then**
        **if** t = 1 **then**
            Freeze $g(.)$
            $\theta_s = \text{copy}(\theta_w)$
        **end if**
        $\mathcal{D}_m \leftarrow (r, y)$
    **end if**
**end for**
**Return:** working model $\theta_w$, and semantic memory $\theta_s$

---

**Table 1.** Parameters for ViT

| Variable | Description |
|---|---|
| patch_size | Patch size of the image fed into the embedding layer |
| hidden_size | Output size of each patch after the embedding layer |
| num_hidden_layers | Number of encoder blocks in the model |
| num_attention_heads | Number of attention heads in the multi-head attention layers |
| intermediate_size | Dimensionality of the intermediate layer in the feedforward layers of MLP |
| hidden_dropout_prob | Dropout probability for the hidden layers |
| attention_probs_dropout_prob | Dropout probability for attention probabilities |
| initializer_range | Range for weight initialization |
| image_size | Size of the input images |
| num_classes | Number of output classes |
| num_channels | Number of input image channels |
| qkv_bias | Whether to include bias in the query, key, and value projections |
| use_faster_attention | Whether to use a faster attention implementation |

**Episodic Memory Architecture –** The episodic memory is updated based on iCarl implementation The representations from the model $g()$ are appended to a list for that particular task during task training. After the end of the task, the representations are sorted based on classes in the tasks. Each class is then sorted based on iCarl herding, which sorts

**Figure 1**. ViT architecture

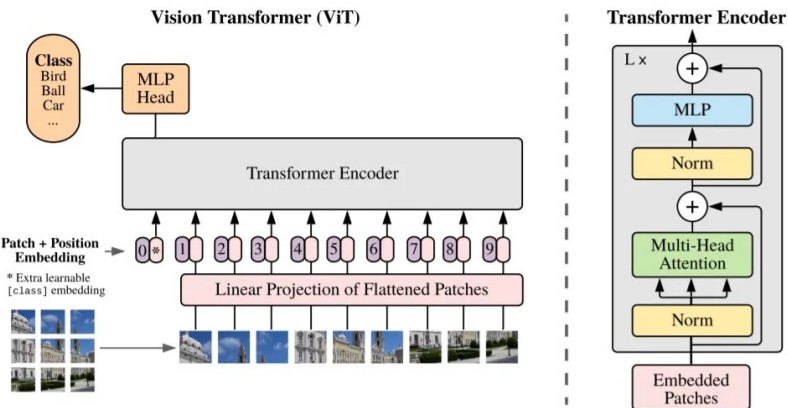

**Table 2**. Parameters for General Training

| Variable | Description |
|---|---|
| base_lr | Hyperparameter for the optimizer |
| weight_decay | Hyperparameter for the optimizer |
| num_classes | Number of classes in the dataset |
| accum_iter | parameter for implementing gradient accumulation, computes gradient and backpropogates after accum_iter number of batches |
| tasks | Number of tasks used for Continual Learning |
| epochs | Number of epochs in training layer |
| batch_size | Number of images processed by the model at a time |
| fine_tune_epoch | Number of epochs for fine tuning after each tasks. |

representations from the most representative representations for that class to the least representative representations of the class.This is done by $select\_exemplar()$ function (Algorithm 2). These sorted representations are then stored in a dictionary corresponding to their class key. The first $n$ representations from each class the model has been trained on until the current task are taken, ensuring that all classes have equal representation in the memory (Algorithm 3). The algorithm for memory updation is presented in Algorithm 4. The algorithm to sample batches during training is presented in Algorithm 5.

**Ideas Implemented** – The ideas implemented from the paper are

- Knowledge consolidation through complementary learning system as illustrated in 1

- Episodic Memory

- Attention Noise $\widetilde{A}$ as illustrated in 5

- Label Noise $\widetilde{T}$

- Supervision Noise $\widetilde{S}$ as described in 7

- The final learning objective which is given by 8 by combining 6, 2.

---

**Algorithm 2** iCaRL CONSTRUCTEXEMPLARSET

---

**Require:** Input: image set $X = \{x_1, \ldots, x_n\}$ of class $y$
**Require:** Input: $m$ target number of exemplars
**Require:** current feature function $\phi : \mathcal{X} \to R^d$
$\quad \mu \leftarrow \frac{1}{n} \sum_{x \in X} \phi(x) \quad$ // current class mean
$\quad$ **for** $k = 1, \ldots, m$ **do**
$\quad\quad p_k \leftarrow \underset{x \in X}{\operatorname{argmin}} \left\| \mu - \frac{1}{k}[\phi(x) + \sum_{j=1}^{k-1} \phi(p_j)] \right\|$
$\quad$ **end for**
$\quad P \leftarrow (p_1, \ldots, p_m)$
$\quad$ Output: exemplar set $P$

---

**Algorithm 3** iCaRL REDUCEEXEMPLARSET

---

**Require:** Input: $m$ $\qquad$ // target number of exemplars
**Require:** Input: $P = (p_1, \ldots, p_{|P|})$ $\qquad$ // current exemplar set
$\quad P \leftarrow (p_1, \ldots, p_m)$ $\qquad$ // keep only first $m$
$\quad$ Output: exemplar set $P$

---

## 4.6 Algorithm Used in this Implementation

Other hyperparameters used in the training implementation specific to BiRT architecture are listed in table 4.6 and the algorithm is illustrated in Algorithm 6

**Table 3.** Hyperparameters specific to BiRT training

| Hyperparameter | Description |
|:---:|:---|
| $\alpha_t$ | Controls amount of label noise |
| $\alpha_a$ | Controls amount of attention noise |
| $\alpha_s$ | Controls amount of trial to trial variability, by applying noise to logits of semantic memory |
| $\alpha_e$ | Controls updation of semantic weights |
| $\alpha_{loss\_rep}$ | Hyperparameter used in representation loss |
| $\rho_{loss\_cr}$ | Hyperparameter used in calculating total loss |
| $\beta_1 loss$ | Hyperparameter used in consistency regulation loss |
| $\beta_2 loss$ | Hyperparameter used in consistency regulation loss |
| $\_gamma$ | Hyperparameter used for updating semantic memory |
| percentage_change | Hyperparameter used in label noise |
| std | Hyperparameter for normal distribution used in creating noise to be applied to semantic memory logits |
| mean | Hyperparameter for normal distribution used in creating noise to be applied to semantic memory logits |
| $c$ | Value of 1 enables attention noise, 0 disables it |

For every tasks, the training loop loops through multiple epochs, for all the batches in that task. It first stores outputs from g() along with each of the labels in a list specific to the tasks. The control variables $\alpha_t\_comp, \alpha_a\_comp, \alpha_s\_comp, \alpha_e\_comp$ are sampled from a normal distribution to control different noises introduced in BiRT. The values of these variables change with every batch. Thus the application of various noises introduced in this paper is random and largely depends on the random values of these control variables which are updated dynamically with every batch and the threshold hyperparameters against which they are compared $\alpha_t, \alpha_a, \alpha_s, \alpha_e$. A minibatch is samples from

---

**Algorithm 4** Update

---

1: **procedure** UPDATE(task_sem_mem_list, task_num)
2:    num_needed_per_class  ←  int(self.max_length/(((task_num + 1) × num_classes)/self.tasks))
3:    self.buffer_images ← []
4:    self.buffer_labels ← []
5:    images_list ← [tup[0] for tup in task_sem_mem_list]
6:    labels_list ← [tup[1] for tup in task_sem_mem_list]
7:    concatenate_images ← torch.cat(images_list, dim = 0)
8:    concatenate_labels ← torch.cat(labels_list, dim = 0)
9:    single_images_list ← torch.split(concatenate_images, 1, dim = 0)
10:    single_images_list ← [tensor.squeeze(dim = 0) for tensor in single_images_list]
11:    single_labels_list ← torch.split(concatenate_labels, 1, dim = 0)
12:    single_labels_list ← [tensor.squeeze(dim = 0).*item*()
13:  for tensor in single_labels_list]
14:    task_set ← set(single_labels_list)
15:    **for** $i$ **in** range(len(single_images_list)) **do**
16:        self.class_separate_list[int(single_labels_list[$i$])].append(single_images_list[$i$])
17:    **end for**
18:    **for** $i$ **in** task_set **do**
19:        select_exemplar_length ← min(self.max_length, len(self.class_separate_list[int($i$)]))
20:        self.class_icarl_list[$i$] ← select_exemplars(self.class_separate_list[int($i$)], select_exemplar_length)
21:    **end for**
22:    **for** $i$ **in** range(num_classes) **do**
23:        **if** ($i$ in self.class_icarl_list) **then**
24:            self.buffer_images.extend(self.class_icarl_list[$i$][0                            : num_needed_per_class])
25:            self.buffer_labels.extend([$i$] × num_needed_per_class)
26:        **end if**
27:    **end for**
28:    self.num_elements ← len(self.buffer_images)
29: **end procedure**

---

the episodic memory only if the memory is not empty, that is after the first task. After which label noise is introduced on it if $\alpha_t\_comp < \alpha_t$, attention noise is implemented if $\alpha_a\_comp < \alpha_a$, noise is added to logits of semantic memory (mini batch sampled from it) if $\alpha_t\_comp < \alpha_t$ and the weights of the semantic model f_s() is updated by interpolating with that of f_w() if $\alpha_e\_comp < \alpha_e$. For the first task, weights of working model f_w() are simply assigned to the semantic model f_s() at the end. Loss is computed and weights of the working model g() and f_w() are updated by the optimizer according to the equations 2, 6 and 8. The memory is updated at the end of each task using iCarl herding [5].

One important thing to notice is that, a copy of the output from the g() after being detached from the compute graph is stored in the episodic memory. Storing without detaching leads to problems in loss computation because it gets backpropagated twice once during training in g() and other time during training in f_w() and f_s().

The training loop implemented in this replication study is summarized in Algorithm 6.

# 5 Tests and Results: CiFAR10

All the Training and Testing was done on V100 GPU with 32GB RAM.

---

**Algorithm 5** Sample Batch

---

    **procedure** GETBATCH
        batch_images ← []
        batch_labels ← []
        **for** $i$ **in** range(self.batch_size) **do**
            index ← np.random.randint(0, self.num_elements)
            batch_images.append(self.buffer_images[$index$])
            batch_labels.append(torch.tensor(self.buffer_labels[$index$]))
        **end for**
        images ← torch.stack(batch_images, dim = 0)
        labels ← torch.stack(batch_labels, dim = 0)
        **return** images, labels
    **end procedure**

---

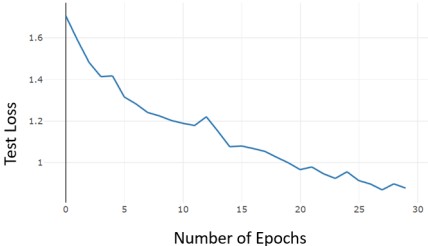

**Figure 2**. Test Loss for ViT trained on CiFAR10

## 5.1 CiFAR10 with ViT Model

The loss and accuracy of ViT model [1] on training dataset after each training epoch is shown in graph **??** 3. The parameters of ViT model are summarized in table . It is to be noted that the dataset was not segregated into tasks for this experiment, and was done on the entire dataset.
The loss criterion is Cross Entropy Loss.
Time taken to train over the entire dataset per epoch is 52 secs.

## 5.2 CiFAR10 with ViT Model separated into $g()$ and $f()$

The loss and accuracy of ViT model separated into $g()$ and $f()$ based on the BiRT paper on training dataset after each training epoch is shown in image 4. The parameters of ViT model are summarized in table 5. It is, again, to be noted that the dataset was not segregated into tasks for this experiment, and was done on the entire dataset. The loss

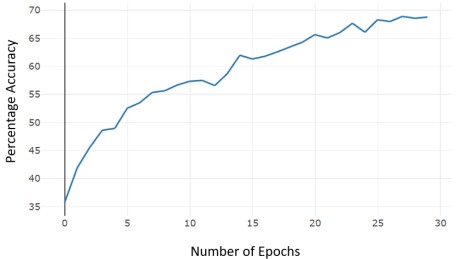

**Figure 3**. Accuarcy Percentage for ViT trained on CiFAR10

---

**Algorithm 6** Training Loop

---

**for** each task **do**
    **for** each epoch **do**
        $train\_loss \leftarrow 0.0$
        $task\_sem\_mem\_list \leftarrow$
        **for** each batch **do**
            $x, y \leftarrow batch$
            $x, y \leftarrow x.to(device), y.to(device)$
            $y\_hat\_temp, \leftarrow model\_g(x, c)$
            $sem\_mem.append((y\_hat\_temp, y))$
            $alpha\_t\_comp, alpha\_a\_comp, alpha\_s\_comp, alpha\_e\_comp \sim U(0,1)$
            **if** $\neg sem\_mem.is\_empty()$ **then**
                $r, r\_y \leftarrow sem\_mem.get\_batch()$
                **if** $alpha\_t\_comp < alpha\_t$ **then**
                    $num\_change \leftarrow int(percentage\_change/100 \times batch\_size)$
                    $indices\_change\_r\_y \leftarrow randperm(len(y))[: num\_change]$
                    $r\_y\_changed \leftarrow randint(0, classes, (num\_change, ))$
                    $r\_y[indices\_change\_r\_y] \leftarrow r\_y\_changed$
                **end if**
                **if** $alpha\_a\_comp < alpha\_a$ **then**
                    $c \leftarrow 1$
                **end if**
                $r\_y\_working, \leftarrow model\_f\_w(r, c)$
                $r\_y\_semantic, \leftarrow model\_f\_s(r, c)$
            **end if**
            $y\_working, \leftarrow model\_f\_w(y\_hat\_temp, c)$
            $y\_semantic, \leftarrow model\_f\_s(y\_hat\_temp, c)$
            **if** $alpha\_s\_comp < alpha\_s$ **then**
                $r\_y\_semantic \leftarrow r\_y\_semantic + noise \sim U(0,1)$
            **end if**
            $loss\_representation \leftarrow 2$
            $loss\_consistency\_reg \leftarrow 6$
            $loss \leftarrow 8$
            $optimizer.\text{zero\_grad}()$
            $loss.\text{backward}()$
            $optimizer.\text{step}()$
            $train\_loss \leftarrow train\_loss + loss.\text{detach}().\text{cpu}().\text{item}()/\text{len}(taskloader)$
            **if** $alpha\_e\_comp < alpha\_e$ and $task\_index > 0$ **then**
                **for** $params1, params2$ in (model_f_s.params, model_f_w.params **do**
                    $interpolated\_params \leftarrow \gamma \times params1.\text{data} + (1 - \gamma) \times params2.\text{data}$
                    $params1.data \leftarrow interpolated\_params$
                **end for**
            **end if**
        **end for**
    **end for**
    **if** $task\_index = 0$ **then**
        **for** $params1, params2$ in model_f_s.params, model_f_w.params **do**
            $interpolated\_params \leftarrow params2.\text{data}$
            $params1. \leftarrow interpolated\_params$
        **end for**
    **end if**
**end for**

---

Table 4. CiFAR10 with ViT Model

| Parameter | Value | Parameter | Value |
|---|---|---|---|
| attention_probs_dropout_prob | 0.0 | qkv_bias | True |
| batch_size | 32 | hidden_dropout_prob | 0.0 |
| epochs | 30 | hidden_size | 48 |
| initializer_range | 0.02 | intermediate_size | 192 |
| lr | 0.0005 | num_attention_heads | 4 |
| num_channels | 3 | num_classes | 10 |
| num_hidden_layers | 4 | patch_size | 4 |
| use_faster_attention | True | learning_rate | 5e-4 |
| weight_decay | 1e-6 | optimizer | Adam |

Figure 4. Test Loss for ViT split into g() and f() trained on CiFAR10

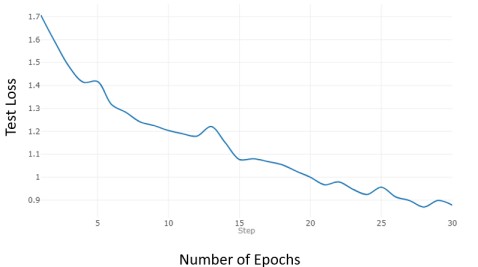

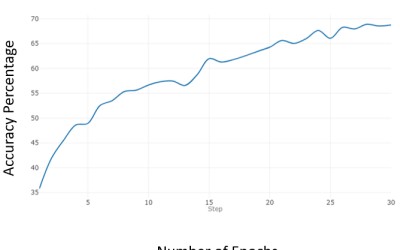

criterion used is Cross Entropy Loss. Time taken to train over the entire dataset per epoch is ~52 secs.

Table 5. CiFAR10 with ViT Model split into g() and f()

| Parameter | Value | Parameter | Value |
|---|---|---|---|
| attention_probs_dropout_prob | 0.0 | qkv_bias | True |
| batch_size | 32 | hidden_dropout_prob | 0.0 |
| epochs | 30 | hidden_size | 48 |
| initializer_range | 0.02 | intermediate_size | 192 |
| lr | 0.0005 | num_attention_heads | 4 |
| num_channels | 3 | num_classes | 10 |
| num_hidden_layers | 4 | patch_size | 4 |
| use_faster_attention | True | learning_rate | 5e-4 |
| weight_decay | 1e-6 | optimizer | Adam |
| c | 0 | | |

## 5.3 CiFAR10 with BiRT architecture for Continual Learning

The parameters used for training CiFAR 10 with BiRT architecture are given in table 6. The 10 classes were divided into 5 tasks with 2 classes each. The architecture was trained for 38 epochs for each task, and then fine tuned on a small balanced dataset with 1000 images for 5 epochs. The average time to update memory after each tasks was 13 mins and the average time to train the model for each epoch per task was 55 sec (36 mins per task for 38 epochs) The values of hyperparameters used in the section is given in 6.

Table 6. Parameters for BiRT Architecture with CiFAR 10

| Parameter | Value | Parameter | Value |
|---|---|---|---|
| attention_probs_dropout_prob | 0.0 | hidden_size | 48 |
| base_lr | 0.0005 | image_size | 32 |
| batch_size | 64 | initializer_range | 0.02 |
| epochs | 20 | intermediate_size | 192 |
| hidden_dropout_prob | 0.0 | num_attention_heads | 12 |
| num_channels | 3 | num_classes | 10 |
| num_hidden_layers | 5 | optimizer | Adam |
| patch_size | 4 | qkv_bias | True |
| tasks | 5 | use_faster_attention | True |
| weight_decay | 1e-06 | accum_iter | 2 |
| $\alpha_t$ | 0.005 | $\alpha_a$ | 0.005 |
| $\alpha_s$ | 0.005 | $\alpha_e$ | 0.003 |
| $\alpha_{\text{loss\_rep}}$ | 0.4 | $\rho_{\text{loss\_cr}}$ | 1 |
| $\beta_{1\,\text{loss}}$ | 0.05 | $\beta_{2\,\text{loss}}$ | 0.01 |
| _gamma | 0.005 | Percentage Change | 5 |
| Std | 1 | Mean | 0 |
| Semantic Memory Length | 500 | | |

**Training BiRT architecture on each tasks successively for 38 epochs –** The loss for each task is summarized in figure 5 and figure 6. The percentage accuracy obtained after training the BiRT architecture for 38 epochs on each tasks was 13.92 %.

**Fine Tuning BiRT architecture for 5 epochs on a balanced dataset of 1000 images –** After tarining on each task, the model was fine tuned on a balanced dataset. The parameters for the ViT model were kept the same as table 6. The loss per epoch is summarized in figure 7. The accuracy obtained after finetuning was 24.96% and the time taken to train the model per epoch was 0.03 sec.

# 6  Test and Results: CiFAR100

The authors trained their BiRT model on 5 tasks of CiFAR 100 for 500 epochs for each task and fine tuned the model for 20 epochs on after each tasks. They report their last accuracy (accuracy on the test dataset of CiFAR 100 after the model has been done training) as 54.15 %. It took them average of 45 mins to train each tasks for 500 epochs.

## 6.1  ViT Model with CiFAR100

The loss and accuracy of ViT model [1] on training dataset after each training epoch is shown in graphs 8 . The parameters of ViT model are summarized in table . It is to be noted that the dataset was not segregated into tasks for this experiment, and was done on the entire dataset.
The loss criterion is Cross Entropy Loss.
Time taken to train over the entire dataset per epoch is ~55 secs.

## 6.2  BiRT Model with CiFAR 100

The BiRT model in this implementation was trained on V100 GPU with 32 GB RAM. The dataset was divided into 5 task each with 20 classes in each. The time taken to update memory after each task was ~5 mins. The time taken to train on each epoch for each task

**Figure 5**. Loss per Task for BiRT on CiFAR10 for 38 epochs for Task 0-2

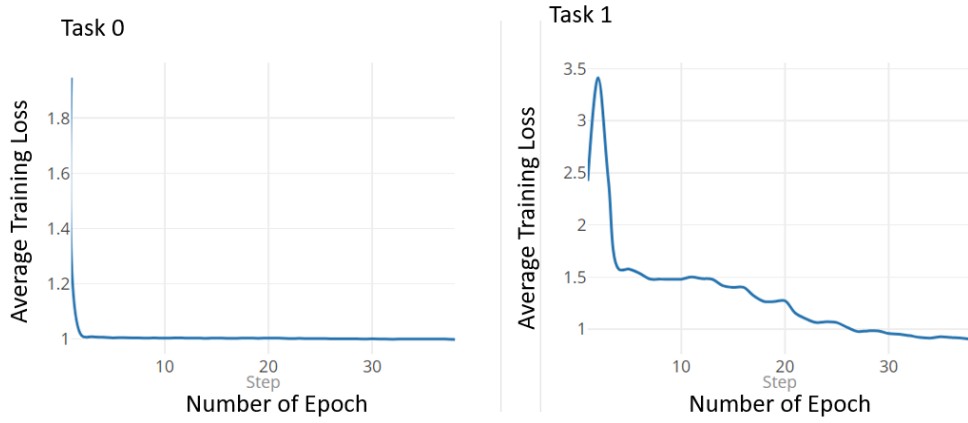

**Figure 6**. Loss per Task for BiRT on CiFAR10 for 38 epochs for Task 3-4

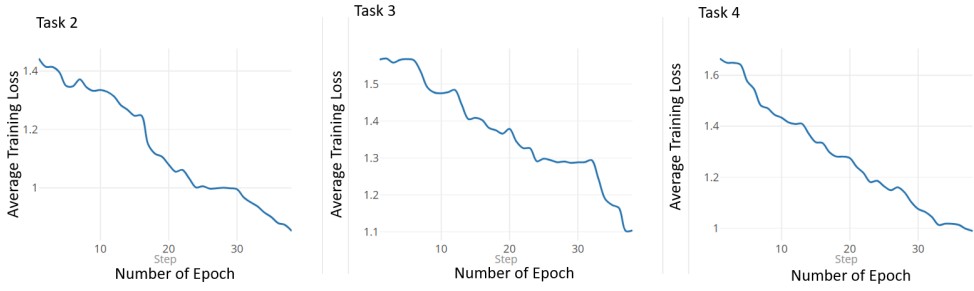

**Figure 7**. Loss per epoch on fine tuning on 1000 CiFAR10 images

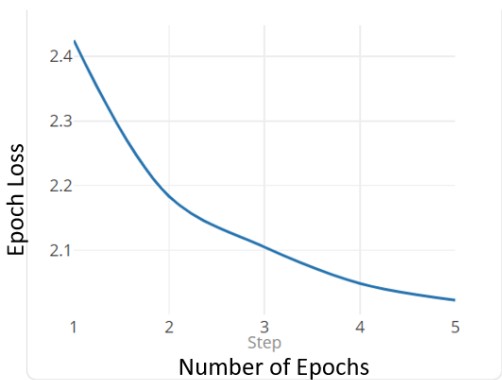

| Name | Value | Name | Value |
|------|-------|------|-------|
| attention_probs_dropout_prob | 0.0 | hidden_dropout_prob | 0.0 |
| batch_size | 32 | hidden_size | 384 |
| epochs | 30 | image_size | 32 |
| optimizer | Adam | initializer_range | 0.02 |
| intermediate_size | 1536 | lr | 0.0005 |
| num_attention_heads | 12 | num_channels | 3 |
| num_classes | 100 | num_hidden_layers | 5 |
| patch_size | 4 | qkv_bias | True |
| use_faster_attention | True | wd | 1e-6 |

**Table 7.** Parameter Values

**Figure 8.** Loss and Accuracy of ViT with CiFAR 100 after 30 epochs

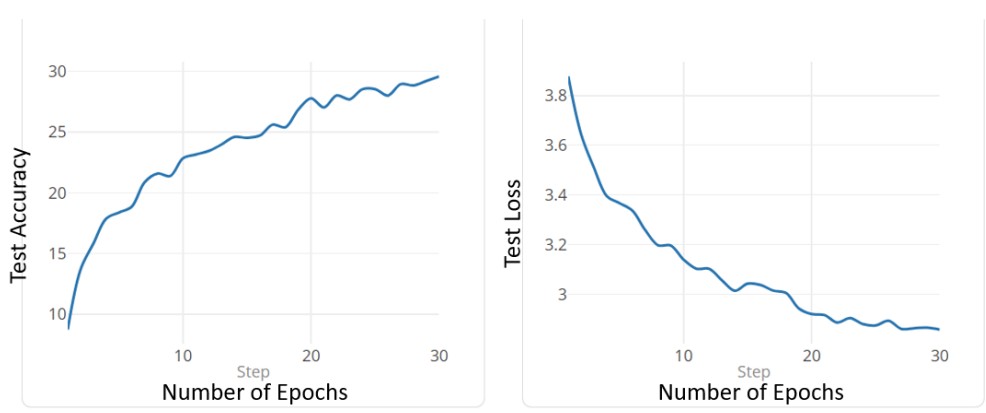

was ~1 min. The model was first trained for 30 epochs each on each of the 5 task and then fine tuned for 5 epochs on a balanced dataset with 1000 images. It was then again trained for 42 epochs on each of the five tasks and then fine tuned again for 5 epochs. The parameters used for the model are given in table 9. The results are summarized below.

**Training BiRT on CiFAR 100 for first 30 epochs** — The average time to train for 1 epoch per task was 60 secs. Thus it took ~30 mins to train for each task. The accuracy obtained after training for 30 epochs on each task was 1.002 %. The loss plots are summarized in 10 and 11

**Fine Tuning BiRT on a balanced dataset of 1000 images from CiFAR 100 for 5 epochs** — The model was then fine tuned on a balanced dataset consisting of 5000 samples for 5 epochs. The accuracy achieved was 4.698%. The loss plot is summarized in the image 12. The average time to train per epoch was 0.02 sec.

**Training BiRT on CiFAR 100 for second 42 epochs** — The average time to train for 1 epoch per task was 60 secs. Thus it took ~43 mins to train each task. The accuracy obtained after

**Figure 9.** Loss and Accuracy of ViT when split into f() and g() with CiFAR 100 after 30 epochs

**Figure 10**. Loss per Task for BiRT on CiFAR100 for 30 epochs for Task 0-1

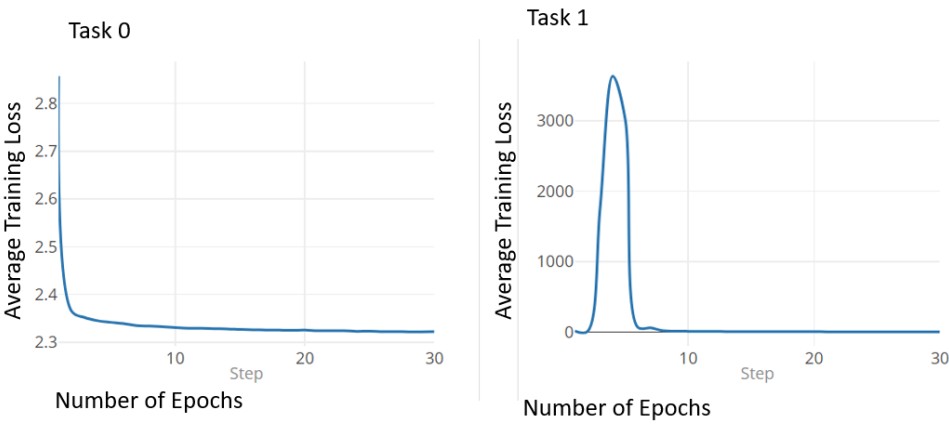

**Figure 11**. Loss per Task for BiRT on CiFAR100 for 30 epochs for Task 2-4

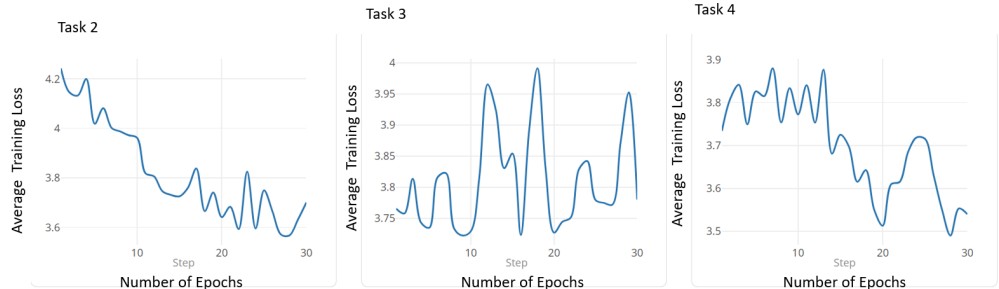

**Figure 12**. BiRT fine tuned on balanced CiFAR 100 dataset of 1000 images

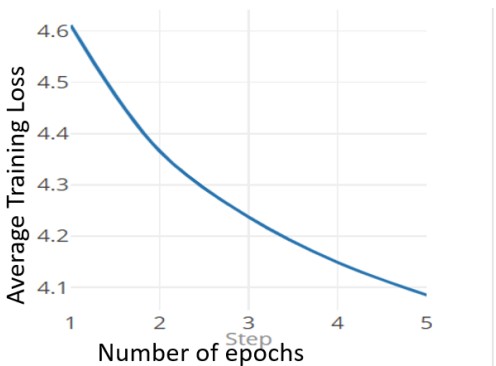

Table 8. Parameter Values for ViT Model with CiFAR 100

| Name | Value | Name | Value |
| --- | --- | --- | --- |
| attention_probs_dropout_prob | 0.0 | hidden_dropout_prob | 0.0 |
| batch_size | 32 | hidden_size | 384 |
| epochs | 30 | image_size | 32 |
| optimizer | Adam | initializer_range | 0.02 |
| intermediate_size | 1536 | lr | 0.0005 |
| num_attention_heads | 12 | num_channels | 3 |
| num_classes | 100 | num_hidden_layers | 5 |
| patch_size | 4 | qkv_bias | True |
| use_faster_attention | True | wd | 1e-6 |

Table 9. Parameters and Hyperparameters for BiRT with CiFAR 100 dataset

| Parameter | Value | Parameter | Value |
| --- | --- | --- | --- |
| Base Learning Rate | 0.0005 | Batch Size | 64 |
| Accum_iter | 2 | Attention Probs Dropout Prob | 0.0 |
| Epochs | 30 | Hidden Dropout Prob | 0.0 |
| Hidden Size | 384 | Image Size | 32 |
| Initializer Range | 0.02 | Intermediate Size | 1536 |
| Num Attention Heads | 12 | Num Channels | 3 |
| Num Classes | 100 | Num Hidden Layers | 5 |
| Optimizer | Adam | Patch Size | 4 |
| QKV Bias | True | Tasks | 5 |
| Use Faster Attention | True | Weight Decay | 1e-06 |
| $\alpha_t$ | 0.005 | $\alpha_a$ | 0.005 |
| $\alpha_s$ | 0.005 | $\alpha_e$ | 0.003 |
| $\alpha_{\text{loss\_rep}}$ | 0.4 | $\rho_{\text{loss\_cr}}$ | 1 |
| $\beta_{1\text{loss}}$ | 0.05 | $\beta_{2\text{loss}}$ | 0.01 |
| $\_gamma$ | 0.005 | Percentage Change | 5 |
| Std | 1 | Mean | 0 |
| Semantic Memory Length | 500 | | |

training for 42 epochs on each task was 3.305 %. The loss plots are summarized in 13 and 14

**Fine Tuning BiRT on a balanced dataset of 1000 images from CiFAR 100 for 5 epochs for the second time** – The model was then fine tuned on a balanced dataset consisting of 5000 samples for 5 epochs. The accuracy achieved was 6.801%. The loss plot is summarized in the image 15. The average time to train per epoch was 0.02 sec.

## 6.3 Conclusion

We first showcase the accuracy of the ViT model used in the BiRT with the dataset for both CIFAR-10 and CIFAR-100. This helps us understand the maximum accuracy achievable by implementing the BiRT training architecture for continual learning when these datasets are divided into different tasks. The key observation is that the reduction in loss is directly proportional to the number of epochs. The rehearsal learning effect of the BiRT algorithm is more evident in a dataset with larger number of classes where the loss takes more epochs to decrease significantly. We hypothesize that this is due to sampling different classes from previous tasks stored in the episodic memory, which

**Figure 13**. Loss per Task for BiRT on CiFAR100 for 42 epochs for Task 0-1

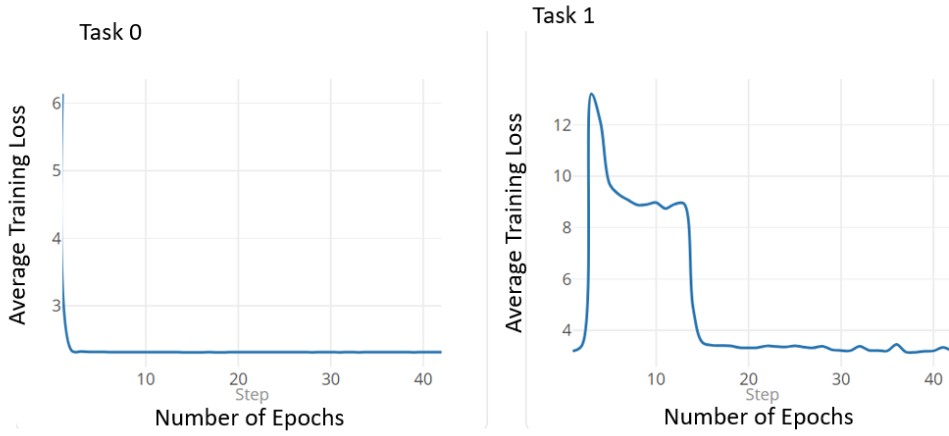

**Figure 14**. Loss per Task for BiRT on CiFAR100 for 42 epochs for Task 2-4

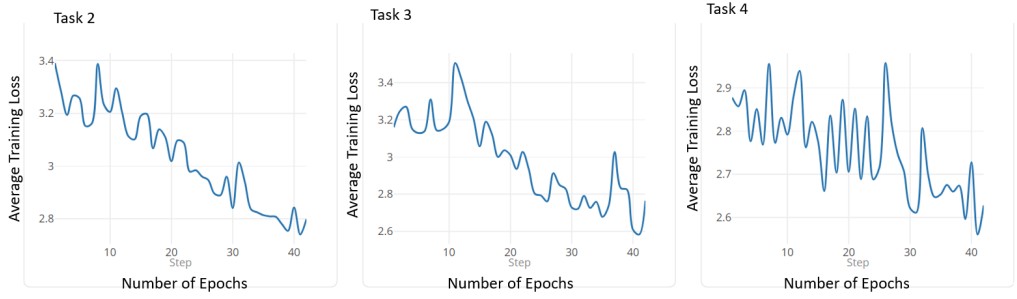

**Figure 15**. BiRT fine tuned on balanced CiFAR 100 dataset

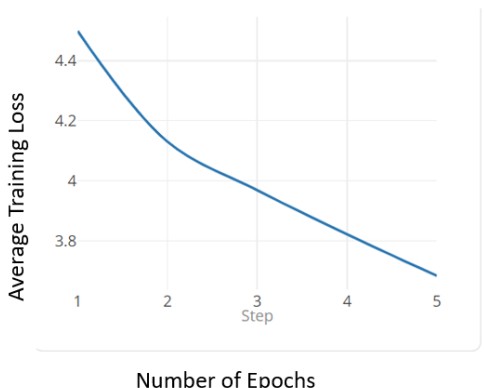

**Figure 16.** BiRT model with CiFAR100 trained in the first 3 task of a total of 5

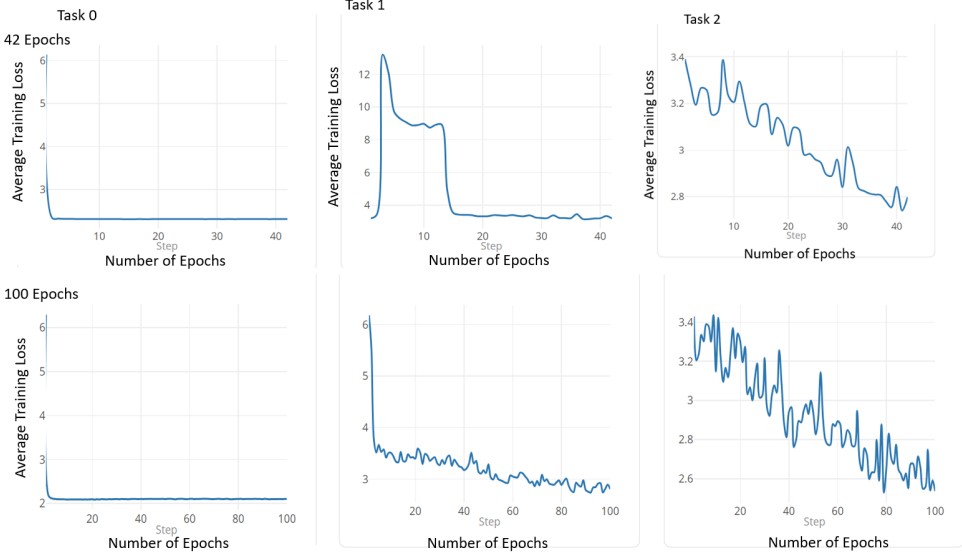

brings in more variation, making it longer for the loss to stabilize and decrease with each successive task. This is evident from figure 16 where CiFAR 100 dataset is divided into 5 task and the model is trained on different number of epochs for the first 3 tasks. Fine-tuning, even on a very small number of images for just a few epochs, has a great effect on the accuracy of the model. We critique that the author should have shown results without fine-tuning as well because in real-world continual learning scenarios, this is not practically possible.

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
