# OpenReview forum: "[Re] BiRT: Bio-inspired Replay in Vision Transformers for Continual Learning"
_purdue.edu/Purdue_University/ML/2023/Hackathon_Reproducibility_Challenge — Purdue University ML 2023 Hackathon Reproducibility Challenge Submission_

### Official Review · Reviewer_ztYq · 2023-12-01
**Review of [Re] BiRT: Bio-inspired Replay in Vision Transformers for Continual Learning**

**Rating:** 8
**Confidence:** 4

**Review:**

The authors of this paper present a thorough investigation into BiRT: Biologically Inspired Replay in Vision Transformers for Continual Learning; an approach that uses biologically inspired mechanisms to obtain an empirical approach towards continual learning. BiRT implements a database of representational class samples along with a snapshot of an older parameter set. This is used to evaluate the performance of the updated model on an task sample that isn't optimized against, to ensure the task-specific loss is minimized in addition to minibatch loss. The authors test BiRT against the ViT architecture to obtain a comparison for the

**Strengths:**
1. This paper explicitly states the assumptions made during the process of replication.
2. The conclusion presents an interesting discussion on the reason for deviations in behaviors observed between the original and the re-implementation. Additional discussions like this would be great!
3. The implementation uses MLFlow to track experimental runs in a well documented manner.
4. Concise technical language used to discuss experimental methodology.
5. Very well detailed documentation of the approach with hyperparameter tables and descriptions. This is a significant improvement over the original paper, that omits the parameters for their experiments.

**Potential Improvements:**
1. A significant portion of the paper is dedicated to explaining the BiRT approach. The focus of the re-implementation papers should be to present a discussion of the shortfalls of the approach, along with ablation studies to either uphold or discredit the original claims made by the paper. While the results for the experiments are presented, it is notably missing a discussion on the results.
2. A large part of the BiRT approach relied on using noise to motivate generalization. It seems like Representation Noise $\tilde{M}$ is not implemented. This seems like a significant divergence from the proposed approach. There needs to include a discussion for why it was omitted from the re-implementation, along with a potential explanation for the behavior of the approach given the change.
3. In the implementation algorithm, LaTeX equations should be used to reduce the size of the variable to their mathematical representation. For example: $alpha\textunderscore t \textunderscore comp \rightarrow \alpha_{t_{comp}}$

**Minor Notes:**
1. Certain components place too much of an emphasis on pseudo-code. As an example, the `toTensor` transformation does not need to be specified, since it performs well.
2. Initially, the paper specifies that the experimental setup was trained for 500 epochs, and in the same paragraph cites time constraints as a reason for not submitting the graph.
3. Section 4.4 can be removed entirely.